# Explainable machine learning model for predicting cesarean section following induction of labor: Development and external validation using real-world data

Yanan Hu[1‡], Xin Zhang[2‡], Valerie Slavin[3,4], Joanne Enticott[1], Emily Callander[1,5]*

**1** Monash Centre for Health Research and Implementation, Faculty of Medicine, Nursing and Health Sciences, Monash University, Melbourne, Victoria, Australia, **2** Department of Electrical and Computer Systems Engineering, Monash University, Melbourne, Victoria, Australia, **3** Gold Coast University Hospital, Gold Coast Hospital and Health Service, Gold Coast, Queensland, Australia, **4** School of Nursing and Midwifery, Griffith University, Gold Coast, Queensland, Australia, **5** School of Public Health, University of Technology Sydney, Sydney, New South Wales, Australia

‡ These authors are co-first authors on this work.
* Emily.Callander@monash.edu

## Abstract

Induction of labor (IOL) is a common yet complex clinical procedure associated with varying risks, including cesarean section (CS). Accurate prediction models may help support more informed, personalized decision-making. This study aimed to develop and validate an explainable machine learning prediction model for CS following IOL. We used population-based administrative perinatal datasets from two Australian states (New South Wales (NSW) and Queensland) covering all births between 2016 and 2019 for model development. Temporal validation was conducted using 2020 births from NSW, and geographical validation using 2016–2018 births from Victoria. We included women with singleton, cephalic, term, live births who attempted IOL and had no prior CS. Seven models (logistic regression, random forest, gradient boosting, LightGBM, XGBoost, CatBoost, and AdaBoost) were developed with hyperparameter tuning and feature selection. Performance was assessed using the area under the receiver operating characteristic curve (AUROC), area under the precision-recall curve, calibration plot (overall and across sociodemographic subgroups), decision curve analysis, Brier Score, and model parsimony. SHAP (SHapley Additive exPlanations) values were used to explain predictor contributions. A total of 180,700 women were included in model development (mean age 31 ± 5 years; CS = 20.8%). The optimal model, developed using XGBoost with ten predictors, achieved AUROCs of 0.76 (95% CI: 0.75–0.77) and 0.75 (95% CI: 0.74–0.76) in temporal (n = 14,527; CS = 22.5%) and geographical (n = 14,755; CS = 19.0%) validations, respectively. The most influential predictors were nulliparity, pre-pregnancy body mass index, and maternal age, while diabetes and hypertension (pre-existing or pregnancy-related) contributed least. Women with higher predicted CS probabilities had increased

**Data availability statement:** De-identified data from this study cannot be shared publicly due to the ethics restrictions imposed by the Research Ethics Committee and the data are owned by third-party organizations. Requests for access to the individual-level data may be made directly to the respective data custodians with appropriate ethics and relevant approvals (please visit: https://www.health.qld.gov.au/hsu/how-to-access-data-from-ssb (Queensland Data Request), https://vahi.vic.gov.au/ourwork/data-linkage/apply (Victorian Data Request), and https://www.cherel.org.au/our-services (New South Wales Data Request)). Detailed codes for data preprocessing, model development and validation, final trained models, web application development, and perinatal data collection manuals are available at https://github.com/Yanan-Hu/CSAI.

**Funding:** YH receives support from the Australian Government Research Training Program (RTP) Scholarship. The funders had no role in study design, data collection and analysis, decision to publish, or preparation of the manuscript.

**Competing interests:** The authors have declared that no competing interests exist.

inpatient costs and maternal morbidity, regardless of actual mode of birth. The final model is accessible via an interactive web application (https://csai-8ccf2690242c.herokuapp.com/). This model demonstrates strong predictive performance using routinely collected maternal factors. Further co-design and implementation research is needed before potential clinical adoption.

## Author summary

An increasing number of pregnant women are having their labor induced. One key concern in this process is the potential need for a cesarean section. To support more personalized and informed decision-making, our study focuses on the first essential step: developing and validating machine learning models that predict the likelihood of cesarean section using routinely collected information available before induction. Leveraging large-scale, real-world data from Australian maternity care, our final model demonstrated strong predictive performance and was designed to be transparent and explainable. We have deployed the best-performing model as a publicly accessible, user-friendly web application (https://csai-8ccf2690242c.herokuapp.com/). This tool provides an individualized prediction and establishes a foundation for future research on clinical implementation, user experience, and real-world impact. Ultimately, it may help guide early treatments, reduce unnecessary obstetric interventions, and improve the efficiency of healthcare resource use.

## Introduction

As a woman's pregnancy approaches term, decisions often need to be made by women with support from clinicians on the mode and timing of birth. This complex decision-making process involves balancing the potential benefits and harms for both women and babies of induction of labor (IOL), pre-labor cesarean section (CS), or awaiting spontaneous onset of labor. One main concern associated with IOL is the potential cascade of interventions, including the likelihood of CS [1]. While prior association studies have extensively explored the relative risks of CS after IOL, either compared with expectant management or spontaneous onset of labor, the absolute, personalized probability of CS cannot be derived from their findings. Importantly, each family brings a unique set of values, preferences, and previous experiences to pregnancy and birth, with different acceptable levels of risk.

Epidemiological studies have also identified various risk factors for CS following IOL [2–5], however, these studies did not shed light on how the presence or absence of each factor (including both risk and protective factors), known before the start of IOL, affects the absolute probability (as opposed to relative risk) of CS in a stepwise and interactive manner. Prediction models are thus needed to explore both the direction and magnitude of each predictor on the predicted probability of CS for future pregnant women.

Recent systematic reviews have identified 23 prediction models of CS following IOL, all used conventional statistical methods (logistic or Bayesian regression) [6,22]. It is necessary to perform a comparative analysis of various methods to assess their difference in performance, as interactions among multiple predictors might not be captured by a single modelling algorithm. Aside from the single modelling algorithm, most prior prediction models included predictors that may not be routinely measured in clinical practice for all women or were known during or after IOL (e.g., duration of IOL or cervical status at the start of IOL). These factors are not predictive for CS before IOL is planned, hence are less valuable for informing women and clinicians prior to planning the labor management. Moreover, most previous studies had a high or unclear methodological risk of bias and poor reporting. These limitations have undermined the reliability and applicability of current models in clinical practice [7].

Our study aimed to fill this evidence gap by developing, temporally and externally validating machine learning models predicting the probability of CS following IOL, using predictors being routinely collected and known before the start of IOL.

## Methods

This study was designed and reported to align with the Transparent Reporting of a multivariable prediction model of Individual Prognosis Or Diagnosis+Artificial Intelligence (TRIPOD+AI) Statement [8].

### Dataset

We utilized population-based retrospective cohorts of women who gave birth in three states: New South Wales (NSW), Queensland (QLD), and Victoria (VIC), which together account for 78% of annual births in Australia. The datasets were sourced from the Perinatal Data Collection (PDC) in each state, a statutory surveillance system covering all live births, and stillbirths of at least 20 weeks' gestation and/or at least 400 grams in weight. Midwives and other birth attendants, using information obtained from mothers and from hospital or other records, completed notification forms for each birth, including labor type and mode of birth. CS was directly defined based on the mode of birth variable in the PDC data dictionary. The administrative perinatal datasets were linked with the hospital admitted patient dataset, which records all inpatient admissions in private and public hospitals.

Women who had singleton, cephalic, live births, at term ($37^{+0}$–$41^{+6}$ weeks of gestation), following attempted IOL were included. Women with a history of CS were excluded. The same eligibility criteria were applied to both training and validation datasets.

All eligible women who gave birth in NSW and QLD between 2016 and 2019 were used for model training; in NSW between January and July 2020 were used for temporal validation; in VIC between 2016 and 2018 were used for geographical external validation.

### Ethics statement

The project received human research ethics approval from the New South Wales Health Service Human Research Ethics Committee (HREC) (HREC/ETH00684/2020.11) on 22 December 2020, and the Australian Institute of Health and Welfare Ethics Committee (EO2020/4/1167) on 29 March 2021. We also received Public Health Act Approval (PHA 20–00684) on 26 February 2021 for a waiver of consent for data collection. No identifiable patient information was provided to the research team.

### Handling of missing data

Due to the small proportion of missing data (< 3%) in some candidate predictors and complete data for the outcome (CS), no data imputation was conducted as recommended in the current literature [9,10]. Instead, missing values were grouped as 'unknown' for categorical variables or assigned an unrealistic outlier value of 99 for continuous variables. Continuous variables were not categorized to avoid potential loss of information. This approach allowed the model to recognize

missingness as potentially informative, while maintaining applicability in real-world settings where user inputs may also be incomplete.

## Model training

Eighteen variables known before the start of IOL and available across the datasets were selected as candidate predictors. No a priori selection based on prior literature or clinical relevance was conducted, as predictor selection was data driven. These included pre-pregnancy body mass index (BMI), maternal age at time of giving birth, woman's country of birth, rurality and socioeconomic status of woman's usual residence, smoking status (before or after 20 weeks), gravidity, parity, diabetes (pre-existing or gestational), hypertension (pre-existing or gestational), weeks of gestation at first antenatal visit or at time of birth, total number of antenatal visits, and birthplace (private or public hospital). Detailed description of candidate predictors is available in Table A in S1 File.

We employed seven classification algorithms to develop our models (detailed in Table B in S1 File): logistic regression, random forest, gradient boosting [11], Light Gradient-Boosting Machine (LightGBM) [12], eXtreme Gradient Boosting (XGBoost) [13], Categorical Boosting (CatBoost) [14], and Adaptive Boosting (AdaBoost) [15]. Logistic regression was included as a baseline for comparison. Hyperparameter tuning (detailed in Table C in S1 File) and predictor selection were conducted within a nested cross-validation pipeline. Finally, all models were retrained using data from all included women. More details of sample size calculation and model training were available in Section A in S1 File.

## Model explainability

SHapley Additive exPlanations (SHAP) beeswarm and waterfall plots were employed to explain the output of our model [16].

## Model performance

Model performance was evaluated using area under the receiver operating characteristic curve (AUROC), area under the precision-recall curve (AUPRC), calibration plot (at population level and across sociodemographic subgroups), and Brier Score, with 95% confidence interval (CI) calculated based on the 2.5th and 97.5th percentiles of the 1,000 bootstrapped values.

## Clinical utility

Decision curve analysis was conducted to evaluate the clinical utility of our models in comparison to default strategies of treating all or no women [17]. This analysis considers the consequences of the decisions made on the basis of our prediction model, which differs from discrimination and calibration measurement. In this context, treatment refers to any action that women or clinicians might consider to improve the health of women with a high predicted CS probability. The specific choice of different treatments is beyond the scope of this study.

## Cost assignment

The cost of each public hospital service event was assigned from the average cost of the Australian Refined Diagnosis Related Group (AR-DRG) reported in the Independent Health and Aged Care Pricing Authority's National Hospital Cost Data Collection (NHCDC) [18] and adjusted in accordance with the adjustments specified in the National Efficiency Price Determination [19]. For admissions in private hospitals, the costs were assigned based on the average cost for each AR-DRG classification identified from the Private Hospital Data Bureau Annual Reports [20]. All costs were inflated based on the Reserve Bank of Australia Inflation Calculator [21] and presented in 2023/24 Australian dollars.

### Interactive figures

Interactive visualizations of all performance measurement figures were created, enabling exploration of the x and y values of each data point (e.g., sensitivity, specificity, and precision) by hovering over and filtering or highlighting specific models by clicking on model names in the legend. Recognizing that every woman may have a different risk tolerance, our models predict individualized probabilities of CS rather than making binary classifications (CS or not). Therefore, classification metrics at a single threshold are not presented. Instead, all combinations of these metrics across the full range of thresholds are made available through the interactive figures. This interactivity enhances clarity and offers a more intuitive way to engage with the underlying data.

SAS V9.4 was used for data pre-processing and Python 3.11 was used for model development, validation, and evaluation.

Table D in S1 File outlines the glossary for terms used in our study, including the technical terminologies used in the fields of medicine, statistics, and machine learning in order to be accessible for a broad readership.

## Results

Table E in S1 File presents the sociodemographic and obstetric characteristics of our training (n = 180,700) and validation cohorts (temporal: n = 14,527; geographical: n = 14,755). Women included in the development dataset had a mean age of 30.7 years, a mean pre-pregnancy BMI of 26.3 kg/m$^2$, with 67.4% born in Australia, 72.7% living in major cities, 53.6% being nulliparous, and 27.6% giving birth in a private hospital. The prevalence of pre-existing diabetes, pre-existing hypertension, gestational diabetes, gestational hypertension, and preeclampsia was 1%, 1.1%, 18.4%, 5.5%, and 2.8%, respectively.

The rate of CS was similar across the training, temporal validation, and geographical validation cohorts (20.8%, 22.5%, and 19%, respectively). The training and temporal validation cohorts shared a comparable distribution of other maternal characteristics. Our geographical validation cohort only consisted of women without any medical conditions or pregnancy complications, resulting in a slightly different distribution.

The final selected model was developed using XGBoost. Although it did not rank first for each performance metric across all datasets, it consistently achieved robust AUROC, AUPRC, and other metrics, while using the fewest predictors (n = 10) and the shortest training time (1 hour) (Fig 1 and Table F in S1 File). This balance of stable performance, parsimony, and computational efficiency enhances its practical value and real-world applicability. The AUROC achieved at 0.757 (95% CI: 0.747–0.765) and 0.747 (95% CI: 0.738–0.755) during temporal and geographical validation, respectively. In decision curve analysis, the XGBoost model had higher clinical utility than 'treat all' and 'treat none' strategies across a reasonable range of threshold probabilities between 0% and 50%. For instance, at a probability threshold of 23%, where the utility of identifying true positives is 3.4 times greater (odds ratio: 77%/23%) than the utility of avoiding unnecessary treatment for false positives, our model is equivalent to identifying 8·9% of women in the temporal cohort who may benefit from early targeted treatment (e.g., closer monitoring) while minimizing unnecessary treatment for false positives, compared to 'treat none' strategy where no women receive treatment.

The top three contributors for higher CS following IOL were nulliparity, higher pre-pregnancy BMI, and older maternal age, whereas the two least contributing predictors were preeclampsia and pre-existing diabetes (Fig 2A). The calculation pathway for a randomly selected woman was displayed in Fig 2B.

In the temporal validation cohort, the model calibrated well across five socioeconomic subgroups (Fig 3). Maternal intrapartum inpatient costs showed a steady increase with predicted probabilities (Fig 4A), with the highest decile incurring nearly double the costs of the lowest decile ($10,341 vs $5,459). Similarly, maternal and neonatal lengths of stay exhibited an increasing trend starting from the 3$^{rd}$ decile (Fig 4C and 4D). In contrast, neonatal inpatient costs during the first month after birth remained relatively stable across deciles, except for the lowest decile, which was associated with lower costs (Fig 4B).

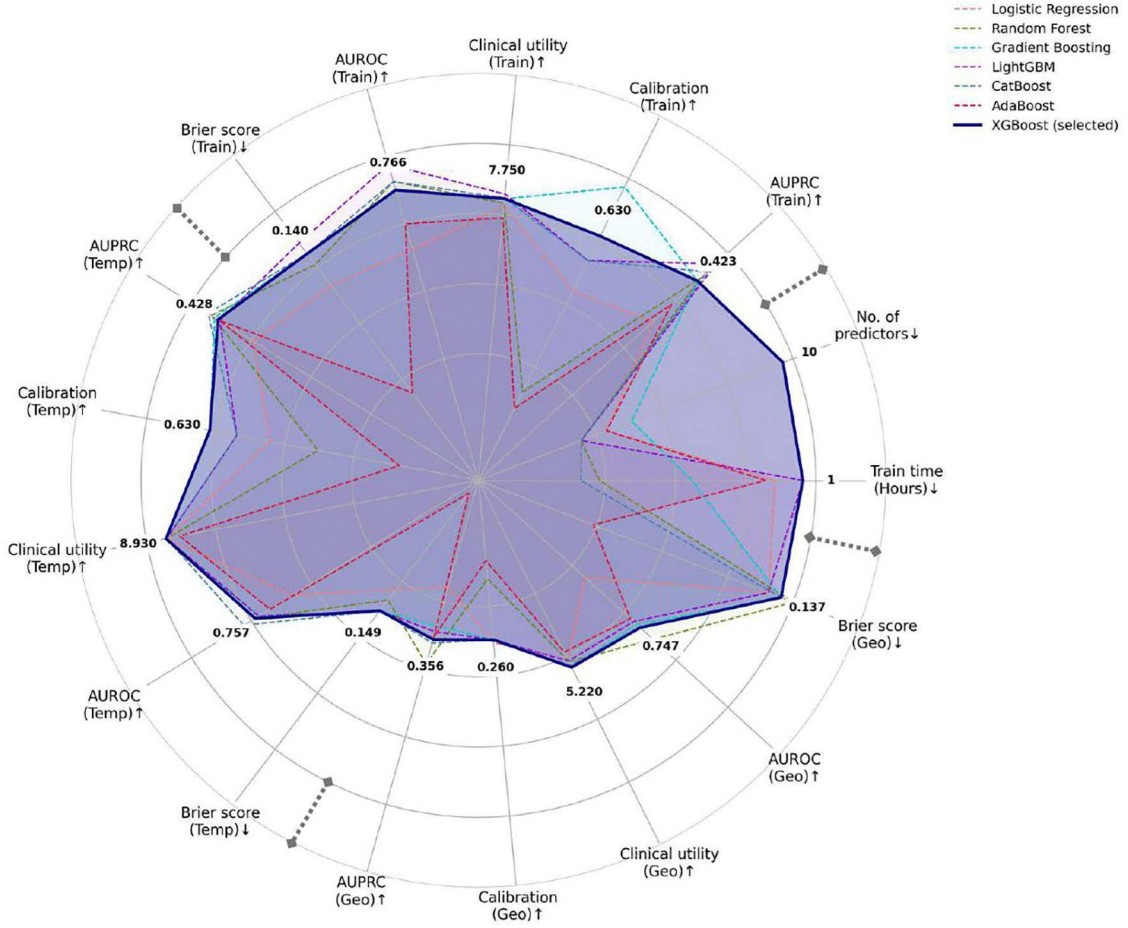

The distance from the center indicates higher performance for each metric, not necessarily higher values. For some metrics, lower values correspond to better performance. The arrows next to each metric label indicate whether a higher value (↑) or a lower value (↓) is preferred for that metric. The value of each performance metric is shown for the XGBoost model. Web links for separate interactive figures and results and for other models are available in Table F in S1 File. Overall training time was based on our system properties: Processor: Intel(R) Xeon(R) Gold 6242 CPU @ 2.80 GHz; Installed RAM: 8.00 GB. The list of included predictors and their importance ranking for each model is available in Fig A in S1 File. Clinical utility was reported at a 23% threshold probability, serving as an illustrative example (a reasonable preference as this is the prevalence of cesarean section following induction of labor across Australia in 2022). The calibration value represents the upper limit of well-calibrated predicted probabilities, where the difference between the observed and predicted probabilities is less than 3%.

Abbreviations: AUROC=Area Under the Receiver Operating Characteristic Curve; AUPRC=Area Under the Precision-Recall Curve; Train=Training cohort; Temp=Temporal validation cohort; Geo=Geographical validation cohort.

**Fig 1. Model performance comparison radar chart.**

**A.**

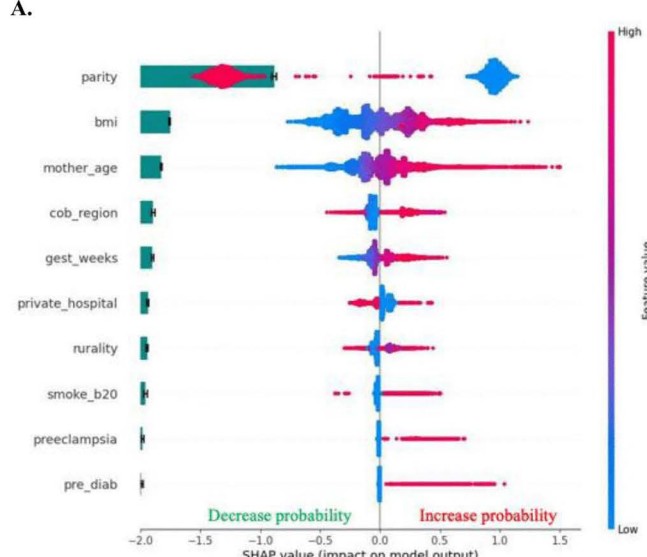

**B.**

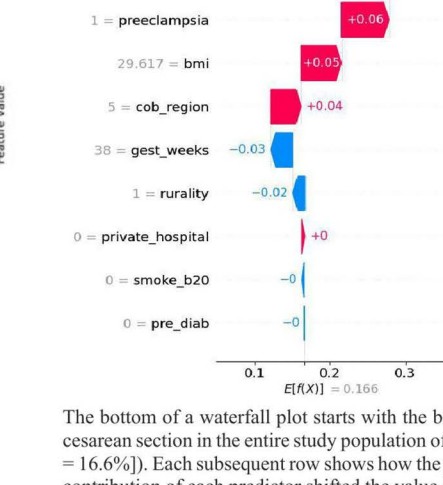

The figure ranked the contribution of predictors based on the mean absolute value of the SHAP values of all women (shown as horizontal green bars, with black error bars indicating 95% CIs derived from 1,000 bootstraps). The horizontal position of each dot (representing a single woman) is determined by the SHAP value of that predictor, with positive SHAP values indicating increased probability of the cesarean section, and dots "pile up" showing density. The color of the dot represents the original value of the predictor (red is higher, blue is lower). From top to bottom in the ranking, red corresponds to parous, higher pre-pregnancy BMI, older maternal age, non-Australian, later weeks of gestation, private hospital, regional/remote area, smoked before 20 weeks, had pre-eclampsia, and had pre-existing diabetes. Definition and response options of each variable are detailed in Table A in S1 File.

The bottom of a waterfall plot starts with the baseline/reference probability of cesarean section in the entire study population of the development dataset (E[f(x) = 16.6%]). Each subsequent row shows how the positive (red) or negative (blue) contribution of each predictor shifted the value from the baseline probability to the predicted probability of cesarean section (f(x) = 54.3%). The grey text before the predictor shows the value of each predictor for the given example.

**Fig 2. The XGBoost model predictors' contribution and effect in the training cohort.**

Distribution of predicted probabilities, composite maternal and neonatal morbidity rate, and graphical abstract are available in Figs B–D in S1 File, respectively. The web links for interactive performance figures are available in Table F in S1 File.

## Discussion

### Main findings

Our study developed, temporally and externally validated, and compared seven prediction models of primary CS at term following IOL, using real-world, population-based administrative perinatal datasets and machine learning algorithms. Our final selected model was developed using the XGBoost classifier with ten routinely collected predictors known before the start of IOL. Along with this superior performance, our model output was explained by quantifying and visualizing the cumulative contribution of each predictor, which gives more holistic and precise information to potential users. Additionally, associated costs were assessed to illustrate the potential cost savings of targeted preventive treatments from the health system perspective. Maternal and neonatal morbidity rates were also evaluated across predicted probability groups, highlighting the association between higher predicted probability and an increased likelihood of adverse outcomes, irrespective of the ultimate actual mode of birth.

The best-performing model was presented as a web application (Fig E in S1 File), named the Caesarean Section After Induction (CSAI) tool—a user-friendly, interactive, and dynamic online calculator—outlining the calculation pathway for

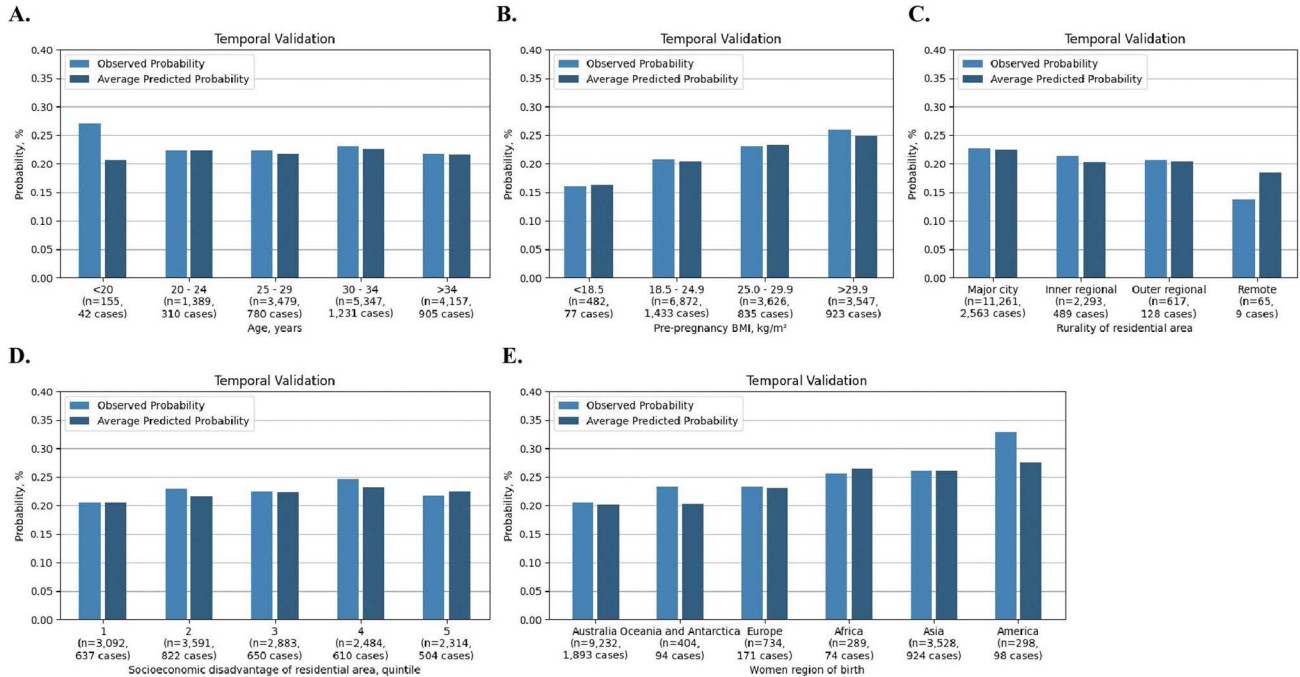

The sample size and the observed number of cesarean sections for each subset are displayed on the x-axis. Missing values are not included for analysis. Socioeconomic status was categorized based on the postcode of usual residence using the Index of Relative Socio-economic Disadvantage (IRSD) in the Socio-Economic Indexes for Areas (SEIFA)—a classification system developed by the Australian Bureau of Statistics (ABS) that ranks geographic areas across Australia according to relative socio-economic advantage and disadvantage. The 1st quintile represents the most disadvantaged areas, while the 5th quintile represents the least disadvantaged areas. Rurality was categorized based on the postcode of usual residence using the Accessibility/Remoteness Index of Australia (ARIA+) used by the Australian Bureau of Statistics (ABS). The 'Remote' category includes both remote and very remote areas. Women's country of birth was categorized based on the Standard Australian Classification of Countries (SACC) used by the Australian Bureau of Statistics (ABS). The 'Australia' category includes external territories and 'Oceania and Antarctica' category excludes Australia. Calibration did not perform well for certain categories with a small sample size (n < 450), such as women aged under 20 years (n = 155). Definition and response options of each variable are detailed in Table A in S1 File.

**Fig 3. The XGBoost model calibration across sociodemographic subgroups in the temporal validation cohort.**

each personalized probability (accessible at: https://csai-8ccf2690242c.herokuapp.com/). S2 File provides a video to demonstrate how potential users can interact with the web application.

### Strengths and limitations

By encompassing all births from all clinical settings, the data reflect the heterogeneity in clinical practice of IOL. Meanwhile, we utilized SHAP plots to provide explainable visualizations of how individual predictor contributes to the final predicted probability of CS, enabling women and clinicians to understand the calculation pathway, thereby enhancing women-specific treatment planning and decision-making. Another strength is our open-source data dictionary, code, and trained models, ensuring transparency and facilitating future research on various topics (e.g., external validation and clinical impact). The models were both temporally and geographically validated using large, heterogeneous, and more recent datasets, proving their robustness and generalizability. Notwithstanding, both the training and validation datasets were collected in Australia, which has a universal health care system. As such, the generalizability to other settings may be restricted, particularly in countries with different structures of health care systems (e.g., the United States and low-resource settings). Moreover, although women's region of birth was included as a predictor, it may obscure more granular socioeconomic and health-related characteristics, such as education level, income, health literacy, cultural or religious practices, and access to care, which could further affect the generalizability of the model.

Our study is also prone to some inherent limitations of historical data. For instance, we proxied the timing of IOL using the weeks of gestation at birth. Whilst a small fraction of women experienced a slow IOL process that lasted more than

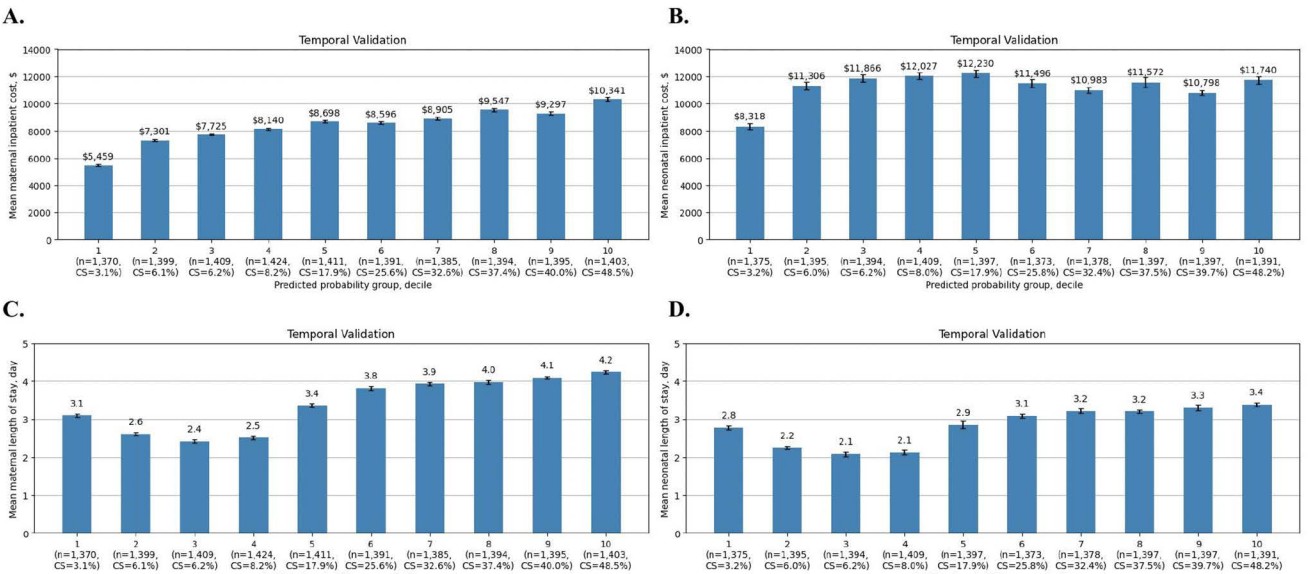

Costs are presented in 2023/24 AUD. The sample size and the observed cesarean section rate for each decile are displayed on the x-axis. The error bar shows the standard error. Only samples with admission records were included in each decile. The 1st decile contains the lowest predicted probabilities, and the 10th decile contains the highest predicted probabilities.
Abbreviation: CS=Cesarean Section.

**Fig 4. Estimated costs by the XGBoost model predicted probability levels in the temporal validation cohort.**

24 hours and consequently gave birth in the next week of gestation, this is unlikely to have a substantial impact on our results. We retrospectively identified women who had gestational hypertension. Future pregnant women might develop this condition immediately preceding or during labor, even though the incidence is quite low. Finally, we were unable to include other predictors that might be routinely collected in other settings (e.g., weight gain during pregnancy or BMI at birth) given their non-availability in our dataset. However, these variables were not the most important predictors in previous models, hence, their exclusion was not expected to significantly influence our model performance.

## Interpretation

In contrast to previous prediction models [6,22], our study investigated alternative modelling methods (i.e., machine learning algorithms), addressed several methodological limitations, including the insufficient sample size with the outcome of interest, selection of predictors based on univariable analysis, and inclusion of predictors being not routinely measured in clinical practice or known only during or after IOL. We also assessed model performance using comprehensive metrics, including fairness and clinical utility. It is essential to point out that only one external validation study examined the clinical utility of four previous prediction models, based on a small dataset (n = 468) [6]. The results of the decision curve should not be used to set the range of threshold probability for the model. Instead, a pre-determined, clinically reasonable range of threshold probabilities should be used to evaluate which model has the highest net benefit across that range [23]. In our case, the reasonable range of threshold probability for IOL against other labor management has not been determined, warranting future qualitative research involving clinicians and women.

Additionally, the heterogeneity of identified predictors in previous prediction models makes external validations and clinical applications challenging. To enhance transparency and facilitate future external validation, we provided a comprehensive data dictionary of our predictors, along with open-source data preprocessing, model development, validation, explanation, evaluation, and final trained models.

Our XGBoost model demonstrated superior robustness compared to most previously temporally or externally validated models, as evidenced by the minimal gap in AUROC between the development and validation cohorts (Table G in S1 File). The narrow range of the 95% CI during bootstrapping further supports our model's low bias and optimism. This could be attributed to the comprehensive nature of our population-based dataset, which lessens selection bias and makes the findings more generalizable to the broader population.

Previously, only Rossi et al have developed and temporally validated their model based on a population-based retrospective cohort, including all live births in the United States between 2012 and 2017 [24]. The AUROC of this model slightly declined from 0.787 in development to 0.780 during temporal validation. Owing to the absence of maternal weight at birth in our datasets, external validation of this model was not feasible. Among all previous prediction models, the model developed by Danilack et al in 2020 has the best discriminative ability, achieving an AUROC of 0.82, 95% CI: 0.79–0.86 on the external cohort [25]. However, the external cohort was a hospital not included in the development cohort, rather than an independent dataset.

Top predictors contributing to a higher probability of CS following IOL in our XGBoost model align with previous models: nulliparity, older maternal age, higher BMI, and later gestational age [6,22]. Of note, in our XGBoost model, both pre-existing and pregnancy-developed diabetes and hypertension are of relatively low contribution. This is likely attributable to their correlation with other key demographic factors, such as BMI, maternal age, and region of birth, which are included in our model and exhibit greater predictive magnitude. This finding is consistent with a previous model [26].

The development and validation of prediction models is just the initial step and needs to be followed by a number of essential procedures before being adopted and implemented into clinical practice. Thus, prospective clinical impact studies, ideally randomized controlled trials, comparing the integration of our CS prediction models into routine maternal care to labor management without the prediction model, were recommended to determine the readiness of our CS prediction models before supporting women's care [27]. Health outcomes, clinical usefulness, cost-effectiveness, and women's experience and satisfaction are all vital outcomes to be measured. For instance, despite the model developed by Levine 2018 [28] having methodological limitations (selected predictors based on univariable analysis and used modified Bishop score as a predictor, which cannot be known early enough before the start of IOL to support decision-making or planning), it has been implemented as a standard of care at one urban academic hospital in the United States. The findings of the implementation reported lower rates of maternal morbidity and CS, and improved women's perception of the quality of care received, without compromising neonatal outcomes [29].

Nonetheless, several challenges remain for the successful clinical implementation of machine learning–based decision support tools [30]. Foremost among these are the limited explainability and transparency of complex algorithms, which constrain trust among both clinicians and women. In addition, the absence of standardized frameworks for evaluating algorithmic bias, fairness, and explainability continues to impede widespread adoption. Integrating prediction models into existing electronic health record (EHR) systems presents further challenges, including compatibility with diverse data infrastructures and secure data exchange. Collectively, these barriers highlight the immaturity of current methodological and implementation science in this domain. Addressing them will require sustained interdisciplinary collaboration encompassing technical innovation, rigorous evaluation, user-centered co-design, ethical and regulatory governance, and policy development, before such models, including ours, can be deployed at scale in a reliable and responsible manner.

## Conclusions

The CSAI prediction tool, using ten routinely collected maternal characteristics, offers objective, individualized, equitable, and explainable probability of CS following IOL. With further co-design and implementation research, it could be used to guide early treatments towards women at high predicted probability, potentially averting unnecessary obstetric interventions and improving the efficiency of healthcare resource use.

## Supporting information

**S1 File. Supplemental material.**
(DOCX)

**S2 File. Demo video.**
(MP4)

## Acknowledgments

The authors acknowledge the following: New South Wales (NSW) Ministry of Health as the source for the NSW Perinatal and NSW Admitted Patients Data Collections; and the Centre for Health Record Linkage (CHeReL) for the provision of data linkage. Queensland Health as the source for the Queensland Perinatal Data Collections; and the Statistical Analysis and Linkage Unit (Queensland Health) for the provision of data linkage. Consultative Councils Unit and the Consultative Council on Obstetric and Paediatric Mortality and Morbidity (CCOPMM) as the source for the Victorian Perinatal Data Collection; and the Centre for Victorian Data Linkage (Victorian Department of Health) for the provision of data linkage. We are grateful to CCOPMM for providing access to the data used for this project and for the assistance of the staff at Safer Care Victoria. The conclusions, findings, opinions, and views or recommendations expressed in this paper are strictly those of the author(s). They do not necessarily reflect those of CCOPMM.

## Author contributions

**Conceptualization:** Yanan Hu, Joanne Enticott, Emily Callander.

**Data curation:** Yanan Hu, Emily Callander.

**Formal analysis:** Yanan Hu.

**Methodology:** Yanan Hu, Xin Zhang.

**Supervision:** Valerie Slavin, Joanne Enticott, Emily Callander.

**Validation:** Yanan Hu.

**Visualization:** Yanan Hu, Xin Zhang.

**Writing – original draft:** Yanan Hu.

**Writing – review & editing:** Xin Zhang, Valerie Slavin, Joanne Enticott, Emily Callander.

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
