## [Decision Letter · Decision Letter 0]

25 Aug 2025

PDIG-D-25-00438Explainable machine learning model for predicting cesarean section following induction of labor: Development and external validation using real-world dataPLOS Digital Health Dear Dr. Hu, Thank you for submitting your manuscript to PLOS Digital Health. After careful consideration, we feel that it has merit but does not fully meet PLOS Digital Health's publication criteria as it currently stands. Therefore, we invite you to submit a revised version of the manuscript that addresses the points raised during the review process. Please submit your revised manuscript within 30 days Sep 24 2025 11:59PM. If you will need more time than this to complete your revisions, please reply to this message or contact the journal office at digitalhealth@plos.org.  Please include the following items when submitting your revised manuscript:* A rebuttal letter that responds to each point raised by the editor and reviewer(s). You should upload this letter as a separate file labeled 'Response to Reviewers '. This file does not need to include responses to any formatting updates and technical items listed in the 'Journal Requirements' section below.* A marked-up copy of your manuscript that highlights changes made to the original version. You should upload this as a separate file labeled 'Revised Manuscript with Track Changes '.* An unmarked version of your revised paper without tracked changes. You should upload this as a separate file labeled 'Manuscript '. If you would like to make changes to your financial disclosure, competing interests statement, or data availability statement, please make these updates within the submission form at the time of resubmission. Guidelines for resubmitting your figure files are available below the reviewer comments at the end of this letter. We look forward to receiving your revised manuscript. Kind regards, Onicio Batista Leal-NetoAcademic EditorPLOS Digital Health Onicio Leal-NetoAcademic EditorPLOS Digital Health Leo Anthony CeliEditor-in-ChiefPLOS Digital Healthorcid.org/0000-0001-6712-6626  **Journal Requirements:**  If the reviewer comments include a recommendation to cite specific previously published works, please review and evaluate these publications to determine whether they are relevant and should be cited. There is no requirement to cite these works unless the editor has indicated otherwise.  **Additional Editor Comments (if provided):****Reviewers' Comments:**  Reviewer's Responses to Questions

**Comments to the Author**

1. Does this manuscript meet PLOS Digital Health’s publication criteria ? Is the manuscript technically sound, and do the data support the conclusions? The manuscript must describe methodologically and ethically rigorous research with conclusions that are appropriately drawn based on the data presented.

Reviewer #1: Yes

Reviewer #2: Yes

Reviewer #3: Yes

2. Has the statistical analysis been performed appropriately and rigorously?

Reviewer #1: Yes

Reviewer #2: Yes

Reviewer #3: Yes

3. Have the authors made all data underlying the findings in their manuscript fully available (please refer to the Data Availability Statement at the start of the manuscript PDF file)?

Reviewer #1: Yes

Reviewer #2: Yes

Reviewer #3: Yes

4. Is the manuscript presented in an intelligible fashion and written in standard English?

Reviewer #1: Yes

Reviewer #2: Yes

Reviewer #3: Yes

5. Review Comments to the Author

Reviewer #1: The authors set out to describe the development of a Clinical Decision Support Tool. In this endevaour the authors proceeded rigorously. As far as the predictors and generalisability of their research goes, the main hinderance would be the usage of region to mask potential socioeconomic and associated health factors. Although these factors are important to consider, they as the authors themself state obscure factors such a gestation diabetes. Their findings such as the risk contribution of nulliparous mothers aligns with previous clinical research not just modelling efforts. Prospective clinical trials will be required for validation and definition of cut-offs to the risk probabilities calculated.

Reviewer #2: Based on my comprehensive review of this manuscript on machine learning models for predicting cesarean section following induction of labor, the statistical analysis demonstrates high technical competence and methodological rigor with several notable strengths but also areas for improvement. The study's greatest strengths lie in its robust design using large, population-based datasets from three Australian states (n=180,700 for development), comprehensive comparison of seven different algorithms from logistic regression to advanced boosting methods, and excellent validation strategy incorporating both temporal (2020 data) and geographical (Victoria) external validation. The authors appropriately employed nested cross-validation with hyperparameter tuning, evaluated performance using multiple metrics (AUROC, AUPRC, calibration plots, Brier score) with proper confidence intervals via bootstrapping, and provided exceptional transparency through open-source code and models. The model's performance remained stable across validation sets (AUROC: 0.757 → 0.747), and the use of SHAP values for explainability enhances clinical interpretability. However, several areas warrant attention: the missing data handling strategy of assigning arbitrary values (99 for continuous variables) rather than principled imputation methods is methodologically suboptimal despite low missingness (<3%); the moderate class imbalance (20.8% cesarean rate) isn't explicitly addressed through appropriate techniques; and the manuscript would benefit from more detailed justification of why XGBoost was superior beyond performance metrics and better discussion of clinical implementation challenges. Additionally, while the comprehensive performance evaluation is excellent, the authors could strengthen their approach by considering interaction terms for key predictors, providing confidence intervals for SHAP values, and conducting sensitivity analyses comparing their missing data approach to proper imputation methods. Despite these areas for enhancement, the work adheres to TRIPOD+AI guidelines, addresses an important clinical question with appropriate statistical techniques, and provides a valuable, practical tool for clinical decision-making. The development of the interactive web application (CSAI tool) represents excellent translation of research into practice, though integration considerations for electronic health record systems could be better addressed. Overall, this represents high-quality research that advances the field of obstetric prediction modeling, with the suggested improvements being refinements rather than fundamental methodological concerns, supporting a recommendation for acceptance with minor revisions.

Reviewer #3: Brief description:

This paper developed and validated an explainable machine learning model to predict the likelihood of cesarean section (CS) following induction of labor (IOL) using large-scale, population-based perinatal data from Australia. They evaluate seven ML algorithms using various performance metrics, including AUROC, AUPRC, calibration plots, decision curve analysis, Brier Score, and model parsimony. Additionally, they identify key predictors using SHAP values. The key predictors include nulliparity, higher pre-pregnancy BMI, and older maternal age. Finally, they provide a public web application offering individualized predictions to support personalized decision-making of IOL.

Suggested improvements:

• Given the CS rate (~20%) and resulting class imbalance, AUROC alone may be insufficient to reflect clinical utility. Please also report F1-score, sensitivity, specificity, and confusion matrices for both temporal and geographical validations.

• According to Table S6, XGBoost is not consistently the top performer in AUROC or AUPRC across datasets. Could you please clarify the decision criteria for selecting XGBoost as the final model?

• Were predictors chosen purely because they were available in the dataset, or based on prior literature and clinical relevance? Please cite supporting references for included variables where possible.

• The manuscript should explicitly describe how the outcome (CS) was ascertained from linked datasets (e.g., CPT codes or other computable phenotypes).

• There are two different front colors used in the text.

6. PLOS authors have the option to publish the peer review history of their article (what does this mean? ). If published, this will include your full peer review and any attached files.

**Do you want your identity to be public for this peer review?** For information about this choice, including consent withdrawal, please see our Privacy Policy .

Reviewer #1: No

Reviewer #2: **Yes: ** Tooba Adil

Reviewer #3: No

---

## [Editor Report · Decision Letter 1]

6 Oct 2025

Explainable machine learning model for predicting cesarean section following induction of labor: Development and external validation using real-world data

PDIG-D-25-00438R1

Dear Dr. Hu,

We're pleased to inform you that your manuscript has been judged scientifically suitable for publication and will be formally accepted for publication once it meets all outstanding technical requirements.

Within one week, you'll receive an e-mail detailing the required amendments. When these have been addressed, you'll receive a formal acceptance letter and your manuscript will be scheduled for publication.

An invoice for payment will follow shortly after the formal acceptance. To ensure an efficient process, please log into Editorial Manager at https://www.editorialmanager.com/pdig/ click the 'Update My Information' link at the top of the page, and double check that your user information is up-to-date. For billing related questions, please contact billing support at https://plos.my.site.com/s/.

Kind regards,

Onicio Batista Leal-Neto

Academic Editor

PLOS Digital Health